# Insertive condom-protected and condomless vaginal sex both have a profound impact on the penile immune correlates of HIV susceptibility

Avid Mohammadi[1]*, Sareh Bagherichimeh[1], Yoojin Choi[2], Azadeh Fazel[1], Elizabeth Tevlin[3], Sanja Huibner[1], Zhongtian Shao[4], David Zuanazzi[4], Jessica L. Prodger[4,5], Sara V. Good[6], Wangari Tharao[3], Rupert Kaul[1,2]*

1 Department of Medicine, University of Toronto, Toronto, Canada, 2 Department of Immunology, University of Toronto, Toronto, Canada, 3 Women's Health in Women's Hands Community Health Center, Toronto, Canada, 4 Department of Microbiology and Immunology, Schulich School of Medicine and Dentistry, Western University, London, Canada, 5 Department of Epidemiology and Biostatistics, Schulich School of Medicine and Dentistry, Western University, London, Canada, 6 Department of Biology, University of Winnipeg, Winnipeg, Canada

* Avid.mohammadi@mail.utoronto.ca (AM); Rupert.kaul@utoronto.ca (RK)

**Data Availability Statement:** The data is shared as  supplementary materials (S2 File).

## Abstract

The penis is the primary site of HIV acquisition in heterosexual men. Elevated penile inflammatory cytokines increase sexual acquisition risk, and topically applied cytokines enhance foreskin HIV susceptibility in an explant model. However, the impact of penile-vaginal sex on these immune parameters is undefined. Heterosexual couples were recruited to the Sex, Couples and Science (SECS) Study, with the collection of penile swabs, semen, cervico-vaginal secretions, and blood after a period of abstinence, and repeated sampling up to 72 hours after either condomless (n = 30) or condom-protected (n = 8) penile-vaginal sex. Soluble immune parameters were quantified by multiplex immunoassay. Co-primary immune endpoints were penile levels of IL-8 and MIG, cytokines previously linked to penile HIV acquisition. One hour after sex there were dramatic increases in penile IL-8 and MIG levels, regardless of condom use, with a gradual return to baseline by 72 hours; similar patterns were observed for other chemoattractant chemokines. Penile cytokine changes were similar in circumcised and uncircumcised men, and repeated measures ANOVA and ANCOVA models demonstrated that the degree of change after condomless sex was explained by cytokine levels in their partners' cervico-vaginal secretions. This may have important implications for the biology of penile HIV acquisition.

## Author summary

In heterosexual men, the penis is the primary site of Human Immunodeficiency Virus (HIV) acquisition. Levels of inflammatory cytokines in the coronal sulcus are associated with an increased HIV risk, and we hypothesized that these may be altered after insertive

**Funding:** This study received funding from Canadian Institutes of Health (CIHR; PJT-156123 and TMI-138656, RK). The funders had no role in study design, data collection and analysis, decision to publish, or preparation of the manuscript.

**Competing interests:** The authors have declared that no competing interests exist.

penile sex. Therefore, we designed the Sex, Couples and Science Study (SECS study) to define the impact of penile-vaginal sex on the penile immune correlates of HIV susceptibility. We found that multiple coronal sulcus cytokines increased dramatically and rapidly after sex, regardless of condom use, with a return to baseline levels by 72 hours. The changes observed after condomless sex were strongly predicted by cytokine concentrations in the vaginal secretions of the female partner, and were similar in circumcised and uncircumcised men. We believe that these findings have important implications for understanding the immunopathogenesis of penile HIV acquisition; in addition, they have important implications for the design of clinical studies of penile HIV acquisition and prevention.

## Introduction

Global HIV incidence remains high, with relatively few prevention strategies targeting heterosexual men. Currently, 17.4 million men are living with HIV worldwide [1], the majority of whom acquired HIV through penile-vaginal sex [2]. While the risk of penile HIV acquisition after condomless insertive vaginal sex with an HIV-infected, antiretroviral-naïve female sexual partner is less than 1/1000, this risk varies considerably based on a number of biological factors [3]. Important determinants of transmission risk from an HIV-infected female partner are the viral load in vaginal secretions and blood, and the presence of bacterial vaginosis (BV) [4–6]. In an HIV-uninfected male partner, risk factors for penile acquisition include the presence of a foreskin, penile microbiome dysbiosis and/or a sexually transmitted infection (STI) [7–10]. The common central pathway through which these biological factors enhance transmission is the induction of inflammation in the genital tract, which causes increased HIV RNA shedding in the vagina and a higher density of HIV target cells in penile tissues, as well as reducing epithelial barrier integrity at both sites [11].

Penile HIV susceptibility is not driven by a single immune parameter, but HIV acquisition in uncircumcised Ugandan men was specifically linked to an elevated level of each of the inflammatory chemokines IL-8 (CXCL8) and MIG (monokine induced by interferon gamma, or CXCL9) in the foreskin prepuce. Both chemokines enhance tissue recruitment of activated immune cells [12], and MIG also enhances HIV susceptibility in cervical tissues [13]. IL-8 is produced by epithelial cells and recruits neutrophils to tissue sites of inflammation [14,15], where the subsequent production of chemoattractants such as MIP-3α recruits highly HIV-susceptible CCR5+CD4+ subsets including Th17 cells and activated CD4+ T cells [16,17]. In keeping with this, levels of IL-8 in the prepuce were directly correlated with the density of Th17 cells and neutrophils in the underlying foreskin tissue [12]. Similarly, levels of inflammatory cytokines/chemokines in the female genital tract (FGT) have been linked with each of HIV acquisition, an increased abundance of neutrophil proteases including MMP9, reduced epithelial barrier integrity and an increase in endocervical CCR5+CD4+ T cell targets [18,19]. Key evidence that inflammatory cytokines at the penile surface play a causal role in HIV acquisition comes from *ex vivo* models, where the topical application of inflammatory cytokine/chemokines (TNF and MIP-1α) to foreskin tissues resulted in the activation of foreskin APCs and more than doubled the density of CD4+ T cell targets in the epithelium of the inner foreskin [20].

The level of inflammatory cytokines/chemokines at the genital surface correlates with HIV acquisition risk in cohorts of both men and women, but genital sampling in these studies was deliberately structured to be at time points quite distinct from sexual activity, while *in vivo*

virus penetration and productive infection of genital tissues occurs rapidly after HIV exposure during unprotected sex [21–23]. Not only does sexual activity have physical effects that may alter genital immunology, but both vaginal fluids and semen are rich in soluble immune factors and have a distinct microbiome [24–27]. This means that during unprotected penile-vaginal sex the penis is exposed to cervicovaginal secretions that contain high levels of inflammatory cytokines such as IL-1α, as well as chemoattractants such as IL-8, MIG, MIP-β and MIP-3α. However, the effects of penile-vaginal sex on penile immunology have not been previously described.

To bridge this gap, the Sex, Couples and Science Study (SECS) protocol recruited established heterosexual couples, assayed baseline genital immune parameters in both the male and female partner, and then defined the effects of penile-vaginal sex on penile immunology. We hypothesized that condomless (but not condom-protected) penile-vaginal sex would lead to transient alterations in penile cytokines, and that these alterations would be more profound and sustained in uncircumcised men as compared to circumcised men.

## Results

### Participant characteristics

Consenting STI-free couples (N = 40) participated in the SECS protocol through the Women's Health in Women's Hands Community Health Center (WHIWH) in Toronto, Canada. Eligible couples abstained from sex for at least 48 hours prior to the baseline sampling visit, as indicated by both self-report and a lack of Prostate Specific Antigen (PSA) in vaginal secretions, and then abstained again for 48 hours prior to engaging in penile-vaginal sex and undergoing longitudinal post-coital sampling. Couples decided which sex group they wanted to participate in, and condom use was confirmed with PSA testing at all study visits. "Condomless sex" was defined based on self-report and the detection of PSA in vaginal secretions one hour after sex. Three couples with vaginal PSA detected at the baseline visit were excluded due to non-abstinence. In addition, three couples with a positive vaginal PSA result at the final study visit were excluded from immune analysis at that time point. One couple participated twice, once in the condomless and once in the condom-protected group, and one couple in the condom-protected group missed their final study visit. Therefore, the final sample set consisted of 37 couples: of these, 30 couples participated in the condomless sex group, and 8 couples in the condom-protected sex group (see S1 Table for comparing baseline characteristics between condomless sex and condom-protected sex group). The median age of the male participants was 22 years (Interquartile range (IQR), 21 to 26 years); 18 male participants were circumcised and 19 uncircumcised. The median duration of the relationship at couple enrolment was 18 months (IQR, 6.5 to 36 months). Demographic data are shown in Table 1.

The median self-reported duration of sexual abstinence prior to the baseline sampling visit was 4 days (IQR, 3 to 7 days), and from baseline sampling to penile-vaginal sex for study purposes was 44 hours (IQR, 41 to 64 hours). After male ejaculation the mean time to collection of the first post-sex genital sample was 75 minutes (IQR, 60 to 90 min; see Fig 1 for the study protocol), to collection of the second sample was 7.2 hours (IQR, 6.8 to 7.5 hours) and to collection of the third sample was 72.8 hours (IQR, 52 to 79.5 hours).

### Penile immune parameters at baseline

Soluble immune parameters that were quantifiable in <60% of penile swabs at the first post-sex visit were analyzed as dichotomous variables (i.e., detectable/undetectable), and those quantifiable in ≥60% of swabs as continuous variables. At baseline, levels of soluble immune factors on the penis demonstrated considerable heterogeneity. Cytokine/chemokines IL-8,

Table 1. Male participant characteristics (N = 37).

| Characteristic | Total (N = 37) |
|---|---|
| **Age (median, range)** | 22 (18–44) |
| **Ethnicity (n)** | |
| Asian | 15 |
| White | 15 |
| Middle Eastern | 3 |
| Latin American | 2 |
| Mixed | 1 |
| **Circumcision status (n)** | |
| Circumcised | 18 |
| Uncircumcised | 19 |
| **Relationship duration (Median, range in months)** | 18 (1–96) |
| **Time since last sex at baseline (Median, range in days)** | 4 (2–23) |
| **Frequency of sex over past month (Median, range)** | 6 (1–25) |
| **Self-reported STI (ever) (n (%))** | 6 (16.2%) |
| **Penile washing (n (%))** | 23 (62.2%) |
| **Female partner bacterial vaginosis* (n (%))** | 5 (13.5%) |

*Defined by Nugent's score

MIG, IL-1α, MIP-1β, E-cadherin and MMP9 were most abundant and considered as continuous variables, while IFNα2a, IL-17, IL-6, and MIP-3α were less frequently detected and therefore analyzed as dichotomous variables.

The majority of soluble immune factors were enriched in the coronal sulcus compared to the penile shaft (S2 Table), particularly E-cadherin (median, 1,167.49; IQR, 318.08 to 3402.88 pg/swab coronal sulcus vs 358.04; IQR, 158.66 to 998.59 pg/swab shaft; p = 0.007; S1 Fig). However, MIP-3α and IL-6 were detected more frequently on the shaft, and IL-1α levels were higher on the shaft (median, 1,154.55; IQR, 526.01 to 2766.80 pg/swab coronal sulcus vs 2,041.04; IQR, 1096.41 to 6070.13 pg/swab shaft; p = 0.047). There was no significant

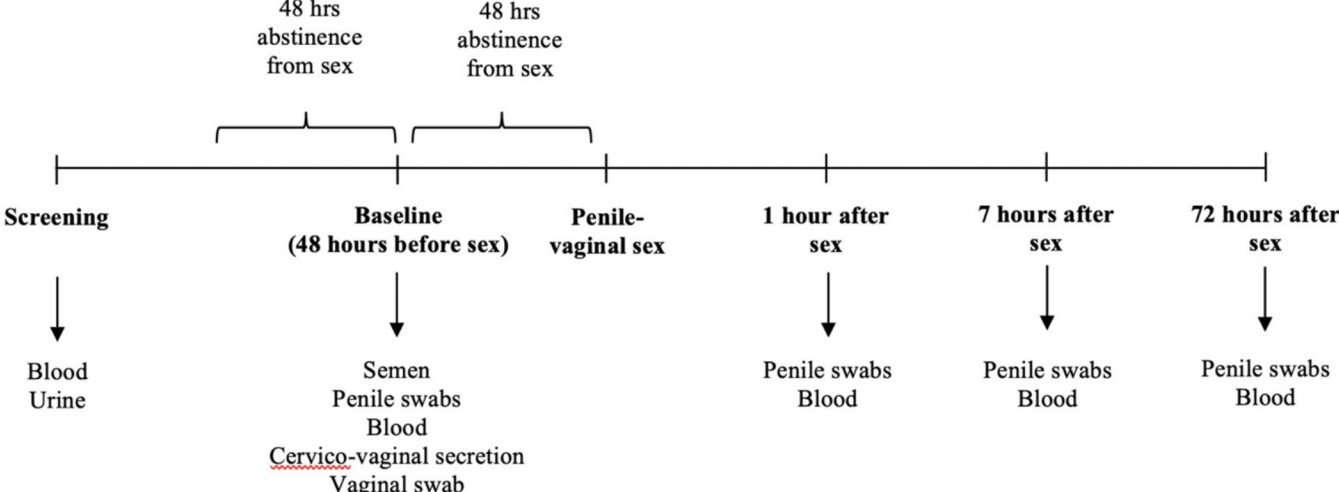

**Fig 1. Overview of the study protocol.** Diagrammatic illustration of the study protocol. Arrows indicate samples collected at specific study visits.

correlation between the baseline level/number of detectable cytokines in the coronal sulcus and participant age, ethnicity, circumcision status, relationship length, condom use, penile washing, time since last sex, number of sex acts during last months and BV status of the female partner. Moreover, no correlation was seen at baseline between penile cytokine/chemokine levels and the Nugent score of a man's female partner.

## Comparison of baseline penile, semen and cervico-vaginal cytokine levels

Since the penile shaft and coronal sulcus have direct contact with cervico-vaginal secretions during condomless penile-vaginal sex, and with semen after ejaculation during condom-protected sex, baseline (ie: pre-sex) levels of immune parameters were compared between these four sites (Table 2). Importantly, while cytokine levels for semen and cervical secretions were corrected for dilution and reported as concentration/unit volume, this was not feasible for swabs of the penile shaft or coronal sulcus, which are therefore reported as concentration/swab. Cervico-vaginal secretions were substantially enriched for IL-8, IL-1α and MMP9, while semen had the highest concentrations of IP-10 and MIG. Cytokine levels at the penile shaft and coronal sulcus were generally much lower (Table 2), with the exception of IL-1α which was enriched on the penile shaft and coronal sulcus compared to semen.

## Impact of penile-vaginal sex on cytokine levels in the coronal sulcus

Levels of IL-8 and MIG in the coronal sulcus immediately after sex constituted our pre-defined primary study endpoints, since their detection at this tissue site has been correlated with HIV acquisition risk [12]. Levels of both chemokines were dramatically elevated in the coronal sulcus one hour after penile-vaginal sex (Fig 2). Increases were most marked after condomless sex (median difference, IL-8 +166.23; IQR, 35.17 to 754.37 pg/swab p<0.001; MIG +4.78; IQR, 0.55 to 32.13 pg/swab, p<0.001; Fig 2A and 2C), but were also apparent after condom-protected sex (median difference, IL-8 +5.59; IQR, 2.55 to 17.02 pg/swab and MIG +6.67 pg/swab; IQR, 1.94 to 42.48 respectively, p = 0.012 for both; Fig 2B and 2D). Coronal sulcus levels of both cytokines subsequently declined over time (Fig 2A, 2B, 2C and 2D). IL-8 levels remained elevated at 7 hours and fell to baseline by 72 hours post sex, while MIG levels returned to baseline by 7 hours, and in the condomless sex group actually fell significantly below baseline levels at 72 hours (median difference, MIG -0.28; IQR, -1.98 to 0 pg/swab, p = 0.001; Fig 2C).

Secondary endpoints included the impact of penile-vaginal sex on coronal sulcus levels of other inflammatory cytokine/chemokines (IP-10, MIP-1β, IL-1α), the epithelial disruption biomarker E-cadherin, and the neutrophil protease MMP9. In general, the impact and time course of changes was similar to that described for our primary endpoints (Fig 2), although the

**Table 2. Baseline immune parameters at different genital sites.**

| Immune parameter | Coronal Sulcus Median pg/swab | Penile shaft Median pg/swab | CVS Median pg/ml | Semen Median pg/ml |
|---|---|---|---|---|
| IL-8 | 2.81 | 1.36 | 28,630.28 | 1,160.55 |
| MIG | 0.34 | 0.09 | 129.02 | 25,283.01 |
| IL-1α | 1,154.55 | 2,041.04 | 25,229.80 | 201.24 |
| MMP-9 | 19.44 | 8.12 | 539,872.01 | 5,590.46 |
| E-cadherin | 1,167.49 | 358.04 | 71,820.66 | 288,695.09 |
| IP-10 | 0.99* | 0.99* | 1,437.75 | 36,334.86 |
| MIP-1β | 13.9* | 13.9* | 472.43 | 646.69 |

* Median value equal to the lower level of detection (LLOD) for immune parameter

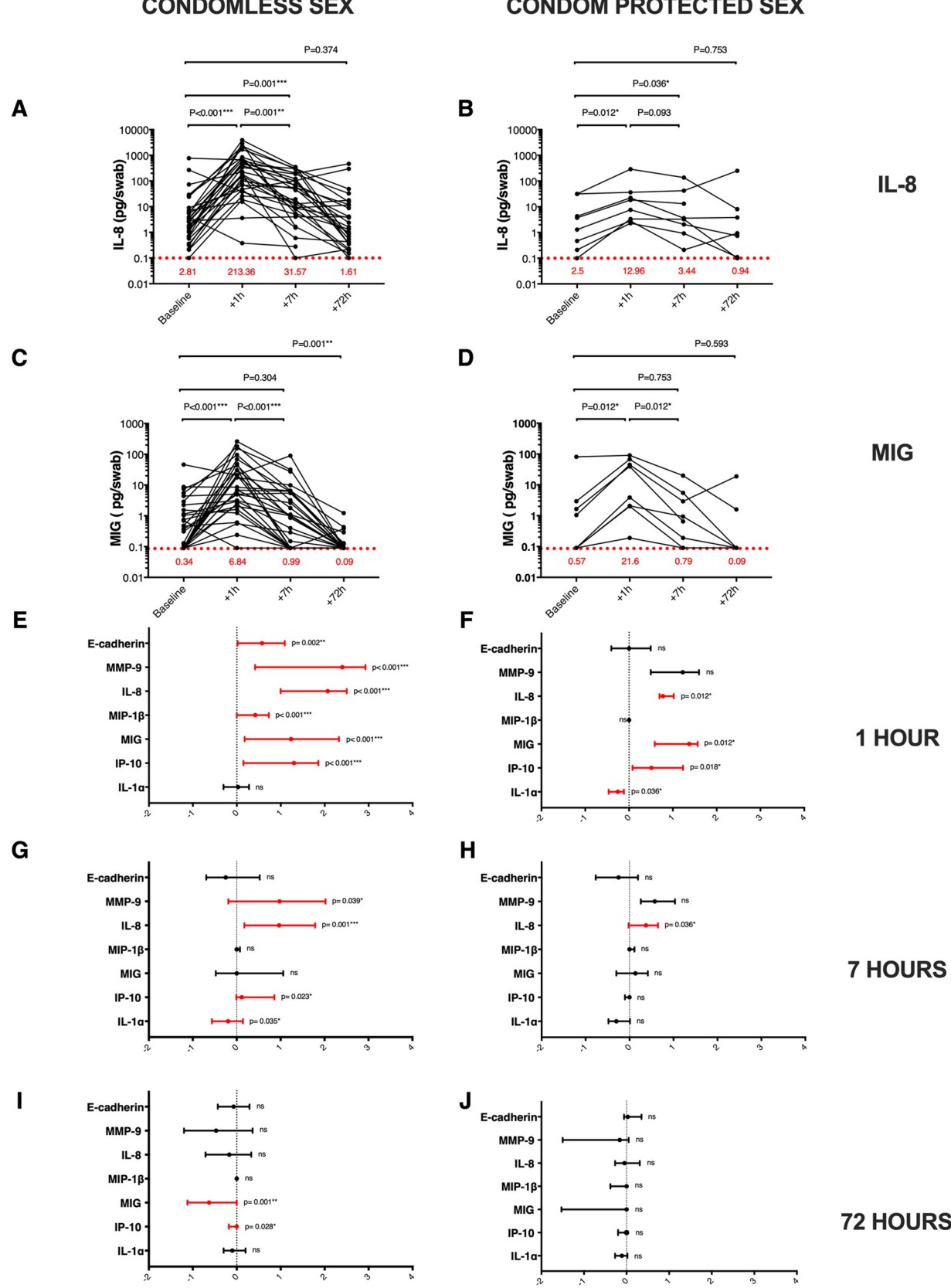

**Fig 2. Impact of penile-vaginal sex on soluble immune factors in the coronal sulcus.** Time course of IL-8 (pg/swab) changes after (A) condomless sex (N = 30), and (B) condom-protected sex (N = 8). Time course of MIG (pg/swab) changes after (C) condomless sex (N = 30), and (D) condom-protected sex (N = 8). Red dotted line represents immune parameter lower level of detection (LLOD), and numbers in red show the median cytokine concentration at each visit. Median change from baseline (log10 transformed) and interquartile range for penile immune parameters are shown at 1 hour post condomless (N = 30; E) or condom-protected sex (N = 8; F), at 7 hours post condomless (N = 30; G) or condom-protected sex (N = 8; H) and at 72 hours post condomless (N = 27; I) or condom-protected sex (N = 7; J). Cytokine concentrations were log10 transformed for illustration. Statistical comparisons used the two-tailed Wilcoxon Signed Ranked test, * p<0.05, ** p<0.01, *** p<0.001.

degree of change varied by cytokine and by condom use. A notable exception was IL-1α, where levels did not change after condomless sex (median difference, + 32.43; IQR, -864.67 to 590.59 pg/swab, p = 0.861; Fig 2E), and actually fell after condom-protected sex (median difference, -1,004; IQR, -2854.46 to -66.03 pg/swab, p = 0.036; Fig 2F). Similarly, cytokines that were present in the coronal sulcus at lower levels and so dichotomized into detectable/undetectable (IL-6, IL-17, MIP-3α and IFNα2a) were generally more commonly detected 1 hour after condomless sex (all p<0.004). There was also an increase in the total number of soluble immune factors detected in the coronal sulcus 1 hour after sex, regardless of condom use (both p<0.032), which fell to baseline by 7 hours and actually decreased significantly by 72 hours in men who had condomless sex (5; IQR, 4 to 6 versus 4; IQR, 4 to 5, p = 0.013). The prior duration of the relationship was not correlated with the extent of penile cytokine/chemokine changes after condomless sex (all p>0.1).

In addition, the fold change in penile cytokines from baseline to 1, 7 and 72 hours post-sex were log2 transformed and presented as a heat-map (Fig 3). Unsupervised hierarchical clustering did not show evidence of clustering by condom presence/absence, although there was a trend for cytokine changes in the condom-protected group to be reduced when compared to their nearest condomless neighbours (S2A Fig).

## The impact of penile-vaginal sex on cytokine levels on the penile shaft

Cytokine changes on the penile shaft generally mirrored patterns seen in the coronal sulcus. All cytokines except IL-1α increased significantly 1 hour after sex (p<0.001) and returned to baseline by 72 hours, with changes being most pronounced in the condomless sex group (S3 Fig). Unexpectedly, among men having condomless sex levels of IL-1α, IL-8, MIG, E-cadherin and MMP9 had all fallen significantly below baseline levels at 72 hours (p<0.05 for all; S3 Fig). The number of detectable cytokines on the penile shaft increased 1 hour after sex, regardless of condom use (both p<0.02), and in men who had condomless sex actually decreased significantly by 72 hours (4.5; IQR, 3.25 to 5.75 versus 4, IQR, 2 to 4, p = 0.001).

## No impact of penile circumcision on penile cytokine changes after penile-vaginal sex

Uncircumcised men are at a higher risk of penile HIV acquisition [28], and so we hypothesized that immune changes after sex would be greater and/or more sustained in uncircumcised men. Among couples who had condomless sex, 16/30 (53%) male partners were uncircumcised. Baseline levels of immune factors in the coronal sulcus did not differ significantly between circumcised and uncircumcised men (S4 Fig). One hour after condomless sex, IL-8 increases were similar in both circumcised and uncircumcised men, and while at 7 hours they only remained significantly elevated in uncircumcised men, the levels were broadly similar (median difference, +41.9; IQR, 5.7 to 150.25 pg/swab versus +6.23; IQR, -0.75 to 141.63 pg/swab p = 0.383; Fig 4A). Unexpectedly, the immediate increase in coronal sulcus MIG levels tended to be more marked in circumcised men vs uncircumcised men (median of difference, +12.1; IQR, 1.12 to 74.10 pg/swab versus +2.06; IQR, 0.23 to 20.82 pg/swab, p = 0.135; Fig 4B),

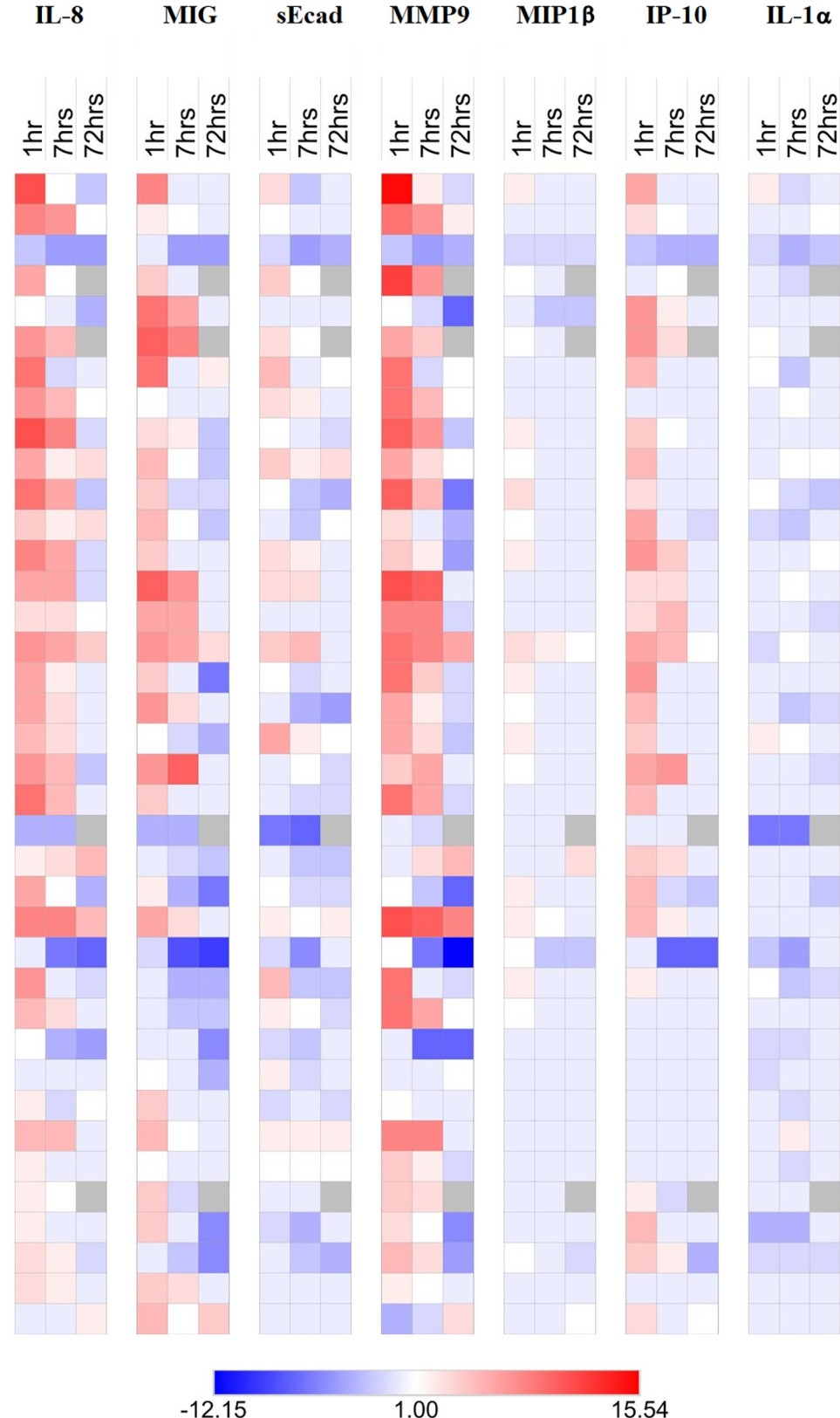

**Fig 3. Penile cytokine changes after sex.** The penile cytokine changes 1 hour, 7 hours and 72 hours after sex (condomless and condom-protected). Penile cytokine changes are presented as log2 transformed fold changes

compared to baseline at each time point after sex. Scale denotes log2 transformed fold changes ranging from -12.15 to 15.54, and 1 is no change. The numbers of participants with paired samples are in 1hr = 38 in 7hrs = 38 and in 72hrs = 34.

with no difference in subsequent declines over time. Similar patterns were seen for circumcised and uncircumcised men in secondary immune endpoints, including coronal sulcus levels of E-cadherin, MMP-9 and other cytokines/chemokines (Fig 4C, 4D, 4E, 4F and 4G), and in the number of detectable cytokines. Moreover, unsupervised hierarchical clustering did not demonstrate clustering based on circumcision status (S2B Fig).

## Penile immune changes reflect exposure to cytokine-rich fluids during sex

Since the baseline concentration of most immune parameters was relatively low in the coronal sulcus (Table 2), we hypothesized that the dramatic changes seen after sex might have been mediated by exposure to cytokine-rich fluids during sex, specifically to cervico-vaginal secretions (CVS) during condomless sex or to ejaculate/pre-ejaculate during condom-protected sex, rather than by the act of sex itself. To explore this hypothesis, we performed a repeated measures ANOVA of the change in coronal sulcus cytokines (IL-8 and MIG) at three time points post-sex (1, 7 and 72 hours) both with and without including the baseline concentration of the cytokine in CVS (for condomless sex) or semen (for condom-protected sex) as a covariate (ANCOVA). This allowed us to test whether differences in the within-person coronal sulcus concentration of cytokines post-sex could be explained by baseline concentrations of IL-8 in the CVS or semen. For all repeated measures ANOVA and ANCOVA models examined, Mauchly's test indicated that the assumption of sphericity was not violated (p>0.05). The repeated measures ANOVA examining changes in coronal sulcus concentration of IL-8 after sex exhibited a significant within person linear decrease over time ($F_{(2, 52)}$ = 47.766, p<0.001), and significant between subject effects ($F_{(1, 26)}$, 123.31, p<0.001), indicating that levels of Il-8 were significantly different among individuals after sex and declined over time. However, when baseline IL-8 in CVS was included in the model as a covariate (ANCOVA), neither the within nor between subject effects were significant ($F_{(2, 48)within}$ = 0.695, p = 0.504; $F_{(1, 24)between}$ = 1.19, p = 0.284, respectively), and there was no interaction between time and baseline IL-8 values ($F_{(2, 48)time*IL-8baseline}$ = 0.357, p = 0.70), indicating that differences in the concentration of IL-8 in the CVS are sufficient to explain the change in IL-8 in the coronal sulcus post-sex, and these likely mediated the penile immune changes.

For the second primary endpoint, coronal sulcus MIG, the repeated measures ANOVA identified a significant linear decline in MIG post-sex in the coronal sulcus but no significant differences between subjects, ($F_{(2, 52) within}$ = 72.19, p< 0.001, $F_{(1, 26) between}$ = 0.007, p = 0.93). However, when female baseline MIG levels in CVS were included as a covariate, neither the within-subject effects nor the interaction between time and MIG baseline levels in CVS were significant ($F_{(2, 48)within}$ = 2.64, P = 0.081; $F_{(2, 48)time*MIGbaseline}$ = 0.206, P = 0.81). These results show that the post-sex levels of coronal sulcus cytokines IL-8 and MIG decline over time, but the overall change in cytokine levels can be explained by exposure to these cytokines in the CVS of their female partners.

Similarly, we employed a repeated measures ANOVA to assess whether adjusting for baseline levels of IL-8 or MIG in semen could explain changes in post-sex levels of coronal sulcus cytokines after condom-protected sex. The repeated measure ANOVA showed significant linear declines for both coronal sulcus IL-8 and MIG (both p< 0.05) and no differences between subjects, but when the values were adjusted for baseline IL-8 in semen, the change was not significant ($F_{(2,10)within}$ = 0.85, P = 0.456), indicating that exposure to IL-8 in self-semen in the

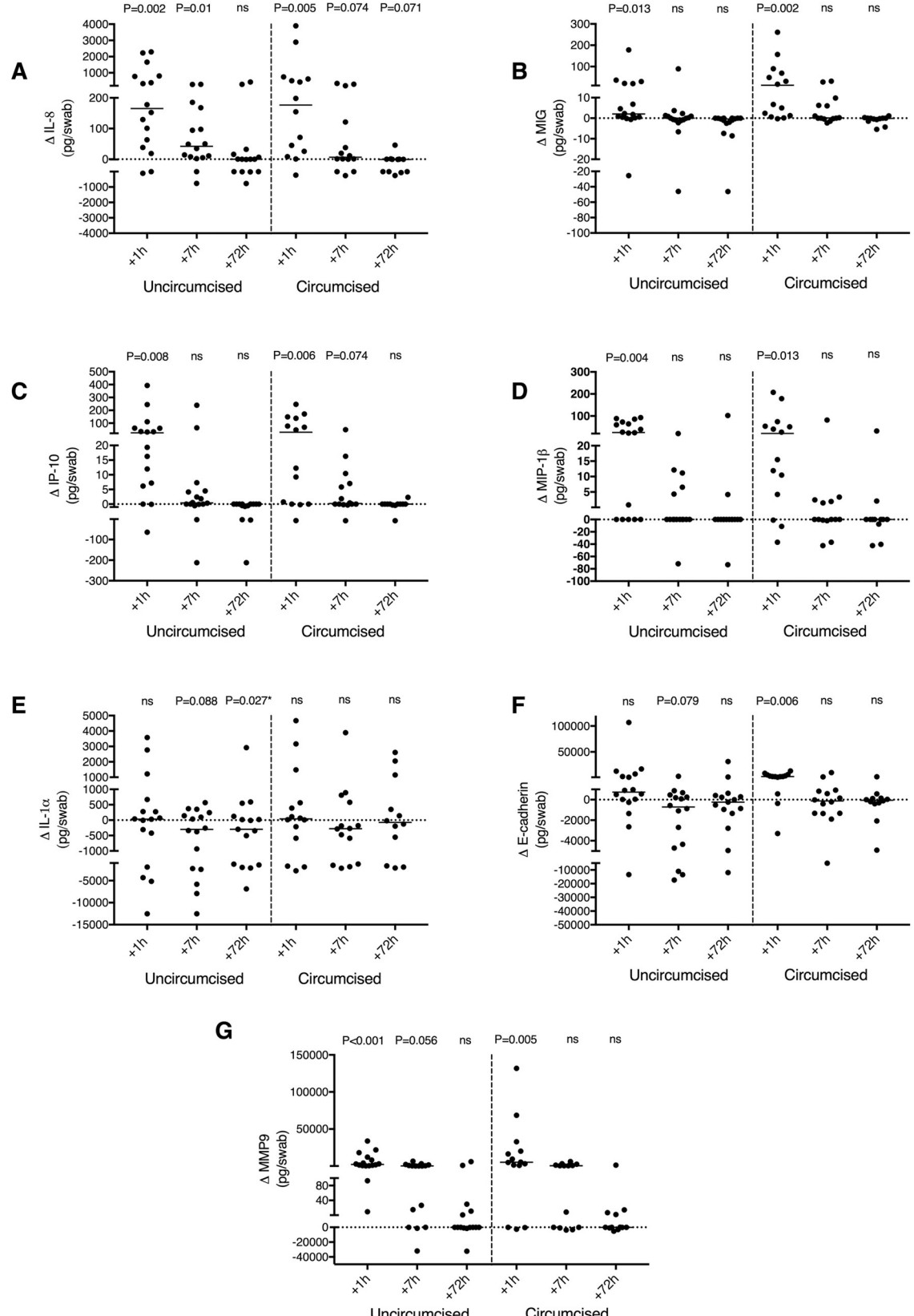

**Fig 4. Circumcision status and penile cytokine changes after condomless sex.** Change (Δ) in the coronal sulcus level of (A) IL-8 (pg/swab) (B) MIG (pg/swab) (C) IP-10 (pg/swab) (D) MIP-1β (pg/swab) (E) IL-1α (pg/swab) (F) E-cadherin (pg/swab), (G) MMP9 (pg/swab) at 1 hour, 7 hours and 72 hours after condomless sex in uncircumcised men (N = 16) and circumcised men (N = 14). P-values refer to change from baseline in immune parameters, using two-tailed Wilcoxon Signed Ranked test. p>0.05 was considered non-significant (ns).

condom protected group can explain the observed changes in coronal sulcus IL-8. On the other hand, correcting for the baseline semen MIG reduced the effect size of within-subject differences in coronal sulcus MIG levels post-sex, but the differences remained significant and there was no interaction between time*baseline MIG ($F_{(2,12)ANOVA}$ = 12.16, P = 0.001 versus $F_{(2,10)ANCOVA}$ = 5.62, P = 0.023). These results suggest that while the changes in coronal sulcus IL-8 concentration post condom-protected sex reflect exposure to semen, additional factors may play a role in coronal sulcus levels of MIG.

## The impact of penile-vaginal sex on T cell parameters in blood

The α4β7 integrin homes lymphocytes from blood to mucosal tissues. Expression of this integrin (and of the proxy CD4+β7 high [29,30]; see S5 Fig for the gating strategy) on blood CD4+ T cells is induced by mucosal procedures such as circumcision and rectal biopsy [31], and an increased frequency of β7 high CD4 cells in the blood is directly linked to heterosexual HIV acquisition [30]. While there was no overall change in the proportion of blood CD4+β7 high T cells after condomless sex (Fig 5, all p>0.2), the proportion of CD4+β7 high T cell increased significantly 7 hours after sex in uncircumcised men (p = 0.024), while in circumcised men there was a significant decrease immediately after sex (p = 0.035). There were no overall changes after condomless sex in the proportion of activated CD4+ T cells, CCR5+CD4+ T cells and CCR6+CD4+ T cells in blood (all p>0.1), although the proportion of Tregs fell at 1 hour and 72 hours (median difference = - 0.48%, p = 0.02; median difference = - 0.38%, p = 0.044, respectively).

## Discussion

Penile-vaginal sex is the major route of HIV acquisition in heterosexual men [2]. While the per-contact risk of HIV acquisition is less than 1/1000 after condomless penile-vaginal sex

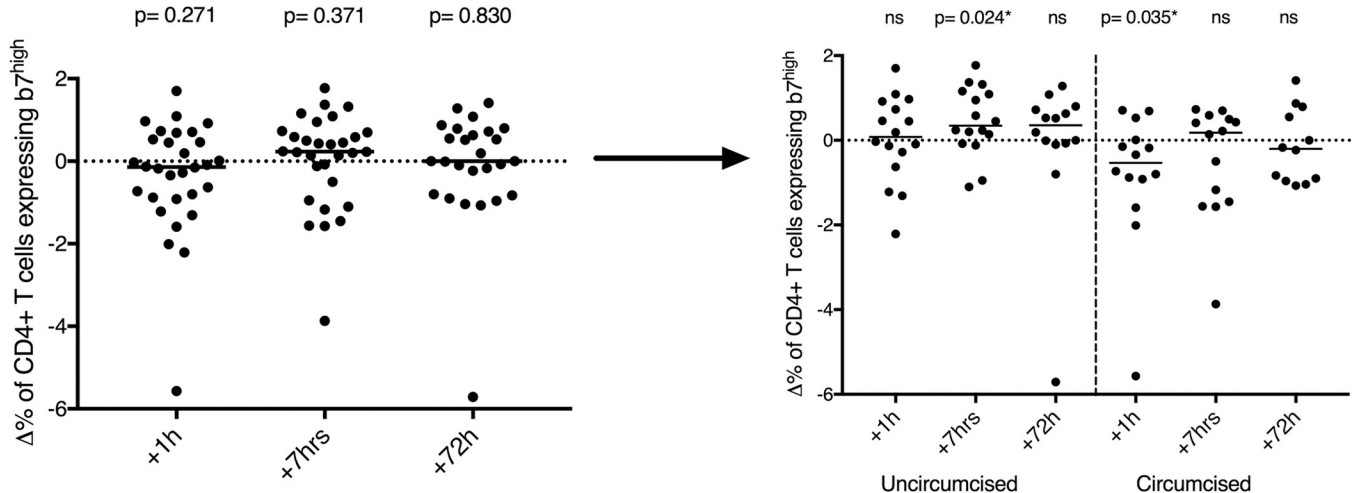

**Fig 5. Impact of penile-vaginal sex on blood CD4+β7 high T cells.** Change (Δ) in the proportion of CD4+β7 high T cell in blood 1 hour, 7 hours and 72 hours after condomless sex (N = 30). Statistical comparisons were performed using two-tailed Wilcoxon Signed Ranked test.

with an antiretroviral treatment-naive female sexual partner who is living with HIV [3], several biological factors can increase this transmission probability [9]. The clearest risk factor for penile virus acquisition is the HIV RNA viral load in the cervico-vaginal secretions of a female partner [4]. However, HIV acquisition in uncircumcised men is also associated with the prior detection of the chemokines IL-8 and MIG in the coronal sulcus, although the origin of these cytokines is unknown and the stability of their levels over time has not been investigated [12]. Since key mucosal events underpinning HIV transmission happen within hours of sexual exposure, the current study was performed to explore in detail the impact of penile-vaginal sex on penile immunology. In general, there were dramatic increases in penile cytokines/chemokines immediately after sex that mirrored the higher levels of these immune factors in the cervico-vaginal secretions of a man's female partner (after condomless sex) or a man's own semen (after condom-protected sex), and that were therefore likely mediated via direct contact with these genital fluids. This was supported by the fact that penile IL-1α levels did not change after condomless sex and actually fell after condom-protected sex, since IL-1α is by far the most abundant cytokine in the coronal sulcus, with levels that approach those in CVS and consistently exceed those in semen (Table 2). All cytokine changes returned to baseline levels within 72 hours with the exception of MIG, whose levels actually fell below baseline 2–3 days after condomless sex.

While penile cytokine levels have been clearly linked to HIV acquisition risk [12], the fact that the dramatic observed changes appeared to reflect "coating" with genital secretions from a female partner might temper the expectation that this would causally affect HIV transmission. However, a previous study showed that the direct topical application of inflammatory cytokine/chemokines (TNF and MIP-1α) to foreskin tissues in an *ex vivo* explant model activated tissue APCs and more than doubled the density of CD4+ T cell targets in the inner foreskin epithelium of the inner foreskin [20], a key determinant of transmission risk [32,33]. In addition, inflammatory cytokines cause epithelial disruption in a model of the female genital epithelium [34]. Therefore, there are at least two mechanisms by which externally-induced changes in cytokine levels on the penile surface could have an important impact on HIV acquisition risk. Interestingly, since bacterial vaginosis (BV) increases inflammatory cytokines in the female genital tract [35,36], our observations also provide a potential explanation for the increased HIV transmission observed to male partners of HIV-infected women with BV, independent of the HIV RNA levels in vaginal secretions [37]. While we did not observe any difference in the coronal sulcus level of IL-1α (a key cytokine increased by BV in the FGT) or in the magnitude of penile IL-1α changes after condomless penile-vaginal sex based on BV status of a man's female partner, we were underpowered to explore this hypothesis since only 5 female partners had BV.

Uncircumcised men have higher levels of IL-8 at the coronal sulcus as compared to circumcised men, and the level of IL-8 at this tissue site correlates with an increased CD4+ target cell density in the underlying tissues and with increased HIV acquisition risk among uncircumcised men [12]. Therefore, we hypothesized that the IL-8 increase after condomless sex would be more pronounced and sustained in the coronal sulcus of uncircumcised men. However, in contrast to our hypothesis, IL-8 (and other cytokine) changes were comparable between circumcised and uncircumcised men, and the timing of cytokine changes was also broadly similar (it should be pointed out that the immune correlates of penile HIV acquisition in circumcised men have not been described). Given that the distal urethra is proposed to be an important site of HIV acquisition in the circumcised penis, future investigation of how condomless sex alters the immune milieu in the urethra would be very interesting.

While the SECS study provides novel insights regarding the impact of penile-vaginal sex on penile immunology, it should be acknowledged that our study has some limitations. While cytokine concentrations in semen and cervico-vaginal secretions could be directly calculated

per unit volume of secretions, the much lower volume of fluid on the penile surface means that cytokine concentrations here could only be reported per swab; since it is not known exactly what fluid volume each penile swab absorbed during sampling, the dilution factor for penile swab samples could not be determined. However, we estimate the average volume of absorbed fluid per swab to be ~50 μl, meaning that cytokine levels would be diluted about 10-fold, and that CVS would still have considerably higher levels of most cytokines tested than the penile coronal sulcus or shaft. This much higher concentration of cytokines in CVS also makes it difficult to ascertain the degree to which local cytokine production in penile epithelial cells might contribute to the changes seen after sex and such endogenous changes, if they occur, might have more immune impact within penile tissues than exogenously applied cytokines. Finally, HIV transmission will generally occur after condomless penile-vaginal sex with an HIV-infected female partner, and for ethical reasons all female partners in the SECS study were HIV-uninfected. However, levels of genital pro-inflammatory cytokines have been found to be comparable between HIV-negative women and women living with HIV [38], which suggests that condomless sex with an HIV-positive female partner would induce similar alterations in penile immunology as observed in our current study.

In summary, we have demonstrated that penile-vaginal sex induced a rapid and dramatic increase in the penile level of cytokines previously linked to HIV acquisition in heterosexual men. These changes generally peaked soon after sex, resolved slowly over 2–3 days, and reflected exposure to cytokine-rich female genital secretions (after condomless sex) and semen (after condom-protected sex). These findings further our understanding of the biology of penile HIV acquisition, and also have implications for the period of abstinence recommended prior to penile sampling for *ex vivo* immune studies.

## Methods

### Ethics statement

The protocol was approved by the HIV Research Ethics Board at the University of Toronto. At the screening visit, the research nurses at WHIWH provided detailed information about the study to potential participants and written informed consent was taken from all interested participants.

### Study design

Couples were recruited into Sex, Couples and Science Study (SECS) protocol from 7/2017 to 11/2018, through the Women's Health in Women's Hands Clinic (WHIWH) (Toronto, Canada). Flyers were posted within the WHIWH centre and across the University of Toronto St. George campus. At the screening visit, participants were tested for sexually transmitted infections and pregnancy. Exclusion criteria were infection with HIV-1/2, syphilis, *Neisseria gonorrhoeae* (GC) and/or *Chlamydia trachomatis* (CT); age <16 yrs; pregnancy; clinical genital ulcers or discharge; irregular bleeding; taking immunosuppressive medications and having taken antibiotics within one month prior to study enrollment. For this observational study, we aimed to recruit 40 couples with an approximately equal number of circumcised and uncircumcised men. The co-primary study endpoints were defined as changes in the coronal sulcus levels of the chemoattractant chemokines, IL-8 and MIG, since these specific chemokines were previously linked to HIV acquisition in heterosexual men [12].

### Sampling protocol

Participants attended five clinic visits; screening, baseline (48 hours before sex) and post-sex follow up visits at 1–2 hours, 7 hours and 72 hours after male partner ejaculation (Fig 1). At

the screening visit, blood and urine were collected for STI diagnostics, and eligible participants were asked to abstain from sex 48 hours before the baseline visit. At the baseline visit, participants completed a demographic/behavioural questionnaire and blood/genital samples were collected. Couples were then asked to abstain from sex again for 48 hours before having one episode of penile-vaginal sex for the SECS study. Couples had either condomless sex or condom-protected sex, based on their preference; men were requested not to wash after sex, and repeat blood/genital samples were collected at 1–2 hours, 7 hours and 72 hours. Men provided samples in the following order: four self-collected penile flocked swabs, urine and blood. Two sets of pre-moistened flocked swabs were collected from the left and right side of the penis, one swab from the coronal sulcus/glans and one swab from the shaft (see S1 File for the sampling instruction). Two swabs were placed in 500 μl PBS, and two swabs were placed in cryovials with no media. All swabs were frozen down at -80˚C within 30 min after collection. At the baseline visit only, semen samples were collected one hour earlier at home by masturbation into a sterile container containing 10 ml RPMI, as previously described [39]. Female participants provided self-collected cervico-vaginal secretions at each visit using an Instead Softcup (Evofem, San Diego, CA) inserted for 1 minute, and a vaginal swab smeared onto a glass slide, that was air-dried for Gram's staining. All genital samples were stored at 4˚C and transported to the laboratory within 30 min of collection. Peripheral blood mononuclear cells (PBMCs) were isolated by Ficoll-Hypaque density centrifugation at 483g for 30 min, counted, and washed in R10 medium. One aliquot of one million PBMCs was used for staining of T cell subsets.

## STI and BV diagnostics

Testing for GC and CT in first-void urine samples was performed by nucleic acid amplification test (NAAT; ProbeTech Assay, BD, Sparks, MD). Testing for HIV-1/2 and syphilis were performed by chemiluminescent microparticle immunoassay (CMIA) (ARCHITECT System, Abbott GmbH & Co. KG). A vaginal swab was smeared onto a glass slide, air-dried and Gram's stained to diagnose bacterial vaginosis (BV) using Nugent criteria. Women with Nugent score > = 7 were considered as BV+. All STI and BV diagnostics were performed at the Sinai Health Systems Microbiology Laboratory, Toronto, Canada.

## PSA testing

Cervical secretions were used to test the presence of Prostate-Specific Antigen (PSA; Serateac PSA Semiquant kit, Göttingen, Germany) as a marker of recent sex.

## Immune cell phenotyping

PBMCs were stained with CD45RA-FITC (BioLegend), CD8- Percp cy5.5 (eBioscience), β7-APC (BD Biosciences), CD127-APCef780 (eBioscience), CD25-BV421(BD Biosciences), CD4-BV650 (BD Biosciences), CCR6-BV711 (BD Biosciences), CD3-BV785 (BD Biosciences), α4-PE (BD Biosciences), CCR5-PE-CF549(BD Biosciences), CCR7-Pe-cy7 (BD Biosciences), HLA-DR-BUV395 (BD Biosciences), CD69-BUV737 (BD Biosciences) and Live/Dead Aqua (Invitrogen). Cells were enumerated using a BD LSR Fortessa X20 flow cytometer (BD Systems) and analyzed with FlowJo 10.4.1 software (TreeStar, Ashland, OR) by the same researcher for consistency.

## Cytokine analysis

All cytokine/chemokine multiplex arrays were performed by research personnel blinded to participant demographics. Penile swabs were vortexed vigorously for 1 minute, the swabs were

inverted and spun down at 112g for 2 minutes. The swabs were discarded and the cryovials were frozen at -80˚C. Cervico-vaginal secretions (CVS) collected by Softcup as described previously (see Methods) were diluted 10-fold using sterile PBS and spun down at 1730g for 10 min. Subsequently, the supernatant was frozen at -80˚C for cytokine analysis. The levels of cytokines IL-1α, IP-10, IL-8, MIP-3α, MIP-1β, IL-17a, IFN-α2a, IL-6, MIG, E-cadherin and MMP9 were measured in duplicate by multiplex immune assay according to the protocol (Meso Scale Discovery, Rockville, MD). The samples were plated at 25 μl per well. The standard curve was used to determine the lower and upper limit of detection and concentration of each analyte (pg/ml). Any sample above the upper limit level of detection was diluted and the multiplex immunoassay was repeated for that sample. The highest LLOD value across runs was selected as the LLOD for each analyte. The LLODs were as follow: IFNα2a = 0.28pg/ml; IL-17 = 1.20pg/ml; MIP-3α = 4.27pg/ml; IL-6 = 0.25pg/ml; IL-1α = 20.6pg/ml; IL-8 = 0.10pg/ml; MIG = 0.086pg/ml; IP-10 = 0.99pg/ml; MIP-1β = 13.9pg/ml; E-cadherin: 53.9pg/ml; MMP-9: 0.35pg/ml. Samples that were below the limit of detection (LLOD) were given the LLOD value. Samples that were above the LLOD value but below LLOD+30% with a high CV were not repeated and the LLOD value was given to those samples. Samples that were above the LLOD with a CV repeatedly higher than 30 were excluded from analysis. Samples provided by each couple at all study visits were run on the same plate to minimize any variability due to plate-to-plate variability.

## Statistics

Data analysis was performed using IBM SPSS v.24 and graphs were prepared by GraphPad Prism v.7. A two-tailed non-parametric statistical analysis was performed to minimize the effect of outliers on our results. A comparison of the baseline cytokine levels was analysed using the Mann-Witney U test. The primary endpoints for cytokine analysis were defined as the levels of immune cell chemoattractant, IL-8 and MIG. Cytokine values were considered detectable, if the value was above LLOD, and undetectable if the value was$< =$LLOD. Penile cytokines that were detectable in $> = 60\%$ of men at 1 hour after sex were considered as continuous variable. To assess correlates of penile cytokine changes after sex, comparisons were performed using the two-tailed Wilcoxon Signed Ranked test. Penile cytokines that were detectable in $<60\%$ of men 1 hour after sex were analysed as dichotomous variables. The McNemar test was used to analyse dichotomous variables. Repeated measures analyses: to analyse the effect on coronal sulcus cytokine levels of exposure to female partner's CVS during condomless sex, or to semen during condom-protected sex, repeated measures ANOVA and ANCOVA were performed on log10 transformed cytokine values in the coronal sulcus 1, 7 and 72 hours post-sex, using the log10 transformed cytokine values from the baseline CVS sample of their partner (condomless sex) or their own semen (condom-protected sex) as a covariate for the ANCOVA. All models were examined to test whether they met Mauchy's test of sphericity. Results of the repeated measure ANOVA and ANCOVA were compared to assess whether adjusting for baseline concentrations of cytokines in the cervico-vaginal cytokines or semen explained within and between subject post-sex changes in coronal sulcus cytokines. For the ANCOVA, the interaction between baseline cytokine concentration * time was also examined to assess if the change in coronal sulcus cytokine levels over time were affected by exposure (baseline levels).

## Supporting information

**S1 Fig. Baseline penile cytokine levels.** Cytokine concentrations (pg/swab) at baseline from swabs of the coronal sulcus and penile shaft (N = 37). Red dotted line represents LLOD for

immune parameter. Statistical comparisons were performed using two-tailed Mann-Witney U test.
(DOCX)

**S2 Fig. Unsupervised hierarchical clustering of penile cytokine changes after sex.** Unsupervised hierarchical clustering was used to visualize the fold change in penile cytokine concentrations 1 hour, 7 hours and 72 hours after sex, based on (A) condom use and (B) penile circumcision status. Scale denotes log2 transformed fold changes ranging from -12.15 to 15.54, and 1 represents no change. The numbers of participants with paired samples are: at 1hr n = 38; at 7hrs n = 38 and at 72hrs n = 34.
(DOCX)

**S3 Fig. Impact of penile-vaginal sex on penile shaft immunology.** Time course of changes of (A) IL-1α (pg/swab) (B) IL-8 (pg/swab) (C) IP-10 (pg/swab) (D) MIG (pg/swab) (E) MIP-1β (pg/swab) (F) E-cadherin (pg/swab), and (G) MMP9 (pg/swab) after condomless sex (N = 30) and condom-protected sex (N = 8). The red dotted line represents immune parameter LLOD, and red numbers the median cytokine concentration at each visit. Statistical comparisons were performed using two-tailed Wilcoxon Signed Ranked test.
(DOCX)

**S4 Fig. Circumcision status and baseline penile immune parameters.** Coronal sulcus cytokine levels (pg/swab) in uncircumcised (N = 16) and circumcised (N = 14) men. The red dotted line represents immune parameter LLOD. Statistical comparisons were performed using two-tailed Mann-Witney U test.
(DOCX)

**S5 Fig. Gating strategy and representative plots for CD4+β7 $^{high}$ T cells in blood.** Cells were gated on lymphocytes, singlets, live, CD3+ cells, CD4+ cells, CD4+T cells expressing β7 $^{high}$.
(DOCX)

**S1 File. Instruction on self-collected penile swabs.**
(DOCX)

**S2 File. The excel file contains data that was used for data analysis and preparing figures.**
(XLSX)

**S1 Table. Comparison of baseline characteristics between condomless and condom-protected group.**
(DOCX)

**S2 Table. Proportion of cytokines detectable in penile swabs at baseline.**
(DOCX)

## Acknowledgments

We acknowledge the time and cooperation of all study participants. We also thank the invaluable support we received from all staff at Women's Health in Women's Hands Community Health Centre who helped us with this project.

## Author Contributions

**Conceptualization:** Avid Mohammadi, Jessica L. Prodger, Sara V. Good, Wangari Tharao, Rupert Kaul.

**Data curation:** Avid Mohammadi.

**Formal analysis:** Avid Mohammadi, Sara V. Good.

**Funding acquisition:** Rupert Kaul.

**Investigation:** Avid Mohammadi, Sareh Bagherichimeh, Yoojin Choi, Sanja Huibner, Zhongtian Shao, David Zuanazzi, Jessica L. Prodger.

**Methodology:** Avid Mohammadi.

**Project administration:** Avid Mohammadi, Sareh Bagherichimeh, Azadeh Fazel, Elizabeth Tevlin.

**Resources:** Rupert Kaul.

**Software:** Avid Mohammadi.

**Supervision:** Wangari Tharao, Rupert Kaul.

**Validation:** Avid Mohammadi.

**Visualization:** Avid Mohammadi.

**Writing – original draft:** Avid Mohammadi, Rupert Kaul.

**Writing – review & editing:** Avid Mohammadi, Sareh Bagherichimeh, Yoojin Choi, Azadeh Fazel, Elizabeth Tevlin, Sanja Huibner, Zhongtian Shao, David Zuanazzi, Jessica L. Prodger, Sara V. Good, Wangari Tharao, Rupert Kaul.

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
