## [Decision Letter · Decision Letter 0]

19 Oct 2021

Dear Mrs Mohammadi,

Thank you very much for submitting your manuscript "Insertive vaginal sex has a profound impact on the penile immune correlates of HIV susceptibility" for consideration at PLOS Pathogens. As with all papers reviewed by the journal, your manuscript was reviewed by members of the editorial board and by several independent reviewers. The reviewers appreciated the attention to an important topic. Based on the reviews, we are likely to accept this manuscript for publication, providing that you modify the manuscript according to the review recommendations.

Please consider comments from reviewer #2 including revising the title e.g. "Insertive condom and condomless vaginal sex increases penile immune correlates of HIV susceptibility" (or similar) and analysing the data in a different way to see if elevated penile IL-8 and MIG levels following sex remain significant. Please address the cell analysis and the results of other markers used and include in the supplementary section and discussion. Some of the figures are too small e.g. Fig 2 and need to be enlarged. Gating strategy (Fig 4) can be moved to the supplementary section. Please consider comments from reviewers aimed to increase clarity of the manuscript.

I have also included additional minor revisions:

Page 5, lines 101- 103. Please provide citation to substantiate statement

Page 19, line 392. RPMI rather than RMPI?

Page 20 line 417. Provide xg force for centrifugation (i.e as done on line 420).

Sincerely,

Gilda Tachedjian, Ph.D.

Associate Editor

PLOS Pathogens

Thomas Hope

Section Editor

PLOS Pathogens

Kasturi Haldar

Editor-in-Chief

PLOS Pathogens

orcid.org/0000-0001-5065-158X

Michael Malim

Editor-in-Chief

PLOS Pathogens

orcid.org/0000-0002-7699-2064

Please consider comments from reviewer #2 including revising the title e.g. "Insertive condom and condomless vaginal sex increases penile immune correlates of HIV susceptibility" (or similar) and analysing the data in a different way to see if elevated penile IL-8 and MIG levels following sex remain significant. Please address the cell analysis and the results of other markers used and include in the supplementary section and discussion. Some of the figures are too small e.g. Fig 2 and need to be enlarged. Gating strategy (Fig 4) can be moved to the supplementary section. Please consider comments from reviewers aimed to increase clarity of the manuscript.

I have also included additional minor revisions:

Page 5, lines 101- 103. Please provide citation to substantiate statement

Page 19, line 392. RPMI rather than RMPI?

Page 20 line 417. Provide xg force for centrifugation (i.e as done on line 420).

Reviewer Comments (if any, and for reference):

Reviewer's Responses to Questions

**Part I - Summary**

Reviewer #1: Here the authors sought to understand how insertive vaginal sex influences the penile mucosal immune environment and subsequent HIV acquisition risk. They performed these analysis by quantification of soluble cytokines by multiplexed ELISA and basic flow cytometric analysis. Further, and importantly, the authors confirmed recent sex (and abstinence) by PSA measurement, which is crucial to minimize the issues with self-reported sex. The use of Multiplexed arrays and flow cytometry is appropriate as soluble cytokines have been strongly associated with HIV acquisition risk and immune cells are crucial in determination for HIV susceptibility. The authors performed flow cytometric analysis with multiple important markers but only report on CD4+B7+ T cells, while not mentioning anything pertaining to the other markers assessed. The authors found that insertive vaginal sex had a pro-inflammatory influence on the penile mucosal environment, regardless of condom use. However, the authors did observe that after condomless sex, the vaginal cytokine concentrations predicted the observed changes in penile cytokines. Interestingly, no difference between circumcision status were observed after condomless sex. Furthermore, these increases in penile cytokines resolved after 72hours. This study is important and novel as it describes the immunological kinetics in the penile mucosal environment after sex, and may provide insight to HIV prevention strategies post sex. Further, as the authors established a clinical cohort to investigate their hypothesis, this cohort may be useful to future studies investigating the influence of sex on the penile mucosal environment and susceptibility to HIV and other infectious diseases.

Reviewer #2: This paper is fairly innovative in attempting to understand the source of cytokines on the penis shaft and glans in circumcised and uncircumcised men; cytokines which have been previously implicated in HIV susceptibility and enhancing in vitro HIV infection. The authors report on a Sex, Couples and Science (SECS) study, which apart from being a sexy name is also no small feat. The co-primary end-points for the study were penile levels of IL-8 and MIG. Regardless of condom use by the men, transient increases in penile IL-8 and MIG levels were observed. The authors conclude, upon measuring these cytokines in the female genital tracts, that “the degree of change after condomless sex was explained by cytokine levels in their partner’s cervico-vaginal secretions.”

Major observations and critique:

1. It is unclear how the authors conclude cytokines are derived from the female cervico-vaginal tract when the levels of IL-8 and MIG show somewhat similar magnitudes and kinetics in men wearing condoms (Fig 2). This should be more carefully explained and currently is unclear.

2. Related to this, it is also unclear why cytokines in the FGT and semen are in a supplementary table, when it is a feature of the conclusions to the paper. This should be considered to placed in the main body of the MS.

3. Results from lines 237-257 on penile changes reflect exposure to cytokine-rich fluids during sex is quite confusing. It is not clear whether the results are linked to a Table or a figure.

4. Lines 280-288 on homing integrin expression.

• Why did the authors look in blood – a clear rationale is missing.

• Why did the authors only show CD4+beta-7 when alpha-4 was used in the flow panel. The authors cite a very old reference from 1993 as some form of justification.

• Is there a reason why the authors used a gut homing marker as opposed to CLA – which would be more appropriate to know which cells in the systemic circulation are migrating to the skin.

• Why did the authors not include CCR5 expression, when it was part of the panel (according to the methods section).

5. The authors may need to change the title of the MS to reflect the data in the paper, as opposed to conjecture, or previous data (albeit from the same group).

6. It would be worth analyzing the cytokines in a different way and looking at fold changes in an unsupervised clustering, PCA and heat-maps; showing how clusters of cytokines may separate out between variables. Do IL-8 and MIG stand up to this approach?

Reviewer #3: This very well written manuscript summarizes findings from a prospective cohort study of heterosexual couples in Canada, aiming to characterize penile immune correlates of penile-vaginal sex. It adds a substantial amount of detail to gaps in the literature through careful and complete measurements of a broad range of relevant measures in both males and females. Comments are minimal – primarily asking for more detail and consideration alternative hypotheses, with no major concerns.

**Part II – Major Issues: Key Experiments Required for Acceptance**

Reviewer #1: na

Reviewer #2: The cytokines should be analyzed in a different way and looking at fold changes in an unsupervised clustering, PCA and heat-maps; showing how clusters of cytokines may separate out between variables. Do IL-8 and MIG stand up to this approach?

Reviewer #3: (No Response)

**Part III – Minor Issues: Editorial and Data Presentation Modifications**

Reviewer #1: 1) Fig 2 should be reworked to enlarge each graph so that the figure can more easily be interpreted. The layout of the figure is fine.

2) Fig 4A the flow cytometry gating strategy should be moved to supplementary data. Some of the data in the supplementary figures, such as S1 and S3 could be moved into main figures.

3) The inclusion of CD4+B7+ cell analysis is important as this subset has been shown to be susceptible to HIV infection. However, even though multiple other important markers were described in the methods, none of these results are shown or eluded too. These should be further discussed, even if no changes were observed and included in supplementary data.

4) Although not crucial to the publication of this manuscript, performing microbial 16s analysis would also greatly improve the scientific rigor of this manuscript and could help explain the observed changes. The authors do mention that they are not powered to assess the impact of female BV on their observed results, but could discuss this in more detail as it is a very important facet of vaginal health.

5) The authors need to include how BV was assessed in their methods section as this is missing.

Reviewer #2: (No Response)

Reviewer #3: Background.

-Nicely written. Importance of IL-8 is described but not for MIG. Suggest to add a sentence or two in paragraph 2 of background, since this is a primary endpoint as stated in Methods.

-Hypotheses clearly stated and supported.

Methods.

- Penile swabbing:

o Could authors please provide more detail on how the penile swabs were self-collected? E.g., were there demonstrations, pictorial guidance, protocol (amount of passes or time), etc. to ensure standardization?

o How much of the shaft was swabbed (e.g., were men directed to swab the whole of it, or maybe a centimeter or so?)

o Could the authors please clarify about samples being “collected left and right side of the penis”? The subsequent part of the sentence says “coronal sulcus/glans” and then “shaft”, so what parts of the penis were “collected from left and right”?

o What type of swabs were used? E.g., flocculated, pre-moistened, etc.

- I don’t recall seeing the Nugent scoring to identify BV (though it says a slide was smeared), but it’s in the footnote of Table 1, but should include it in Methods.

- As there was such a smaller n of condom protected sex, how was participation in the “condomless sex group” vs. “condom-protected sex group”. Presumably, participants self-selected?

Results

- Table 1.

o Age – “18-44” is likely range. Please add (range) after Age in the Characteristic column. Same for other continuous variables.

o Presumably continuous variables are presented as medians? Or maybe they’re means. Please specify in the table.

o Why are some variables reported as % (e.g., self-reported STI, penile washing, female partner BV) while others are reported as n? Needs to be consistent throughout.

o How was penile washing defined? Was this primarily among uncircumcised men?

- Figure 1 is nice. Figure 2 and 3 graphs are useful, but very tiny for this reviewer; I had to go to 200%. Hopefully journal can represent that better in publication or maybe most other readers have larger monitors. Or better eyesight.

- The baseline data showing no differences by any covariates should be added to the Supplementary files.

- Lines 230-231: “unexpected the immediate increase in coronal sulcus MIG levels tended to be more marked in circumcised men vs.” (presumably uncircumcised men).

- It’s a small sample size with multiple comparisons.

Discussion

- Is it known how penile cytokines change/don’t change after masturbation? This would be an additionally interesting control.

- If these changes occur regardless of condom use – by exposure to cervicovaginal secretions if condomless sex or by exposure to man’s own semen after condom protected use – could this also in part be a physiological response, such as after exercise which also induces IL-8? Authors posit that the increases “were therefore likely mediated via direct contact with these genital fluids”; but the cervicovaginal fluids vs. semen likely have different “things” in them eliciting response?

- Median duration of relationship was 18 months, with IQR 6.5 to 36 months. With so few having "newer" relationship, is it possible that the “impact” of sexual activity attenuates over time? Reviewer understands the sample size is small and that “new” relationship vs. more established, but could this perhaps contribute to lack of differences in some expected areas?

- Lines 325-335 on why no difference observed by circumcision status: could the authors please add some details to “future studies” "would be interesting" so that the sentence has more meaning/contribution?

- What might be happening with cytokine/chemokine response for those who have penile-vaginal sex ongoing/several times per week? Would that provide different explanatory information with regards to HIV risk?

PLOS authors have the option to publish the peer review history of their article (what does this mean?). If published, this will include your full peer review and any attached files.

Reviewer #1: No

Reviewer #2: **Yes: **Clive M Gray

Reviewer #3: No

Figure Files:

Data Requirements:

Reproducibility:

References:

---

## [Decision Letter · Decision Letter 1]

13 Dec 2021

Dear Mrs Mohammadi,

We are pleased to inform you that your manuscript 'Insertive condom-protected and condomless vaginal sex both have a profound impact on the penile immune correlates of HIV susceptibility' has been provisionally accepted for publication in PLOS Pathogens.

Best regards,

Gilda Tachedjian, Ph.D.

Associate Editor

PLOS Pathogens

Thomas Hope

Section Editor

PLOS Pathogens

Kasturi Haldar

Editor-in-Chief

PLOS Pathogens

orcid.org/0000-0001-5065-158X

Michael Malim

Editor-in-Chief

PLOS Pathogens

orcid.org/0000-0002-7699-2064

Reviewer Comments (if any, and for reference):

Reviewer's Responses to Questions

**Part I - Summary**

Reviewer #2: All reviewer comments have been addressed

**Part II – Major Issues: Key Experiments Required for Acceptance**

Reviewer #2: (No Response)

**Part III – Minor Issues: Editorial and Data Presentation Modifications**

Reviewer #2: (No Response)

PLOS authors have the option to publish the peer review history of their article (what does this mean?). If published, this will include your full peer review and any attached files.

Reviewer #2: **Yes: **Clive M Gray

---

## [Editor Report · Acceptance letter]

29 Dec 2021

Dear Mrs Mohammadi,

We are delighted to inform you that your manuscript, "Insertive condom-protected and condomless vaginal sex both have a profound impact on the penile immune correlates of HIV susceptibility," has been formally accepted for publication in PLOS Pathogens.

Best regards,

Kasturi Haldar

Editor-in-Chief

PLOS Pathogens

orcid.org/0000-0001-5065-158X

Michael Malim

Editor-in-Chief

PLOS Pathogens

orcid.org/0000-0002-7699-2064